# Synthesized Differentiable Programs

## Abstract

Program synthesis algorithms produce interpretable and generalizable code that captures input data but are not directly amenable to continuous optimization using gradient descent. In theory, any program can be represented in a Turing complete neural network model, which implies that it is possible to compile syntactic programs into the weights of a neural network by using a technique known as *neural compilation*. This paper presents a combined algorithm for synthesizing syntactic programs, compiling them into the weights of a neural network, and then tuning the resulting model. This paper's experiments establish that program synthesis, neural compilation, and differentiable optimization together form an efficient algorithm for inducing abstract algorithmic structure and a corresponding local set of desirable complex programs

## 1 Introduction

Program synthesis efficiently induces abstract computer programs from data. Alternatively, gradient optimization induces parameterized functions which can be seen as a relaxed form of program search [1, 2, 3]. However, programs recovered via gradient optimization will be represented as real-valued weights, in contrast to code in a higher-level language. Generally, program synthesis is more appropriate for finding abstract algorithmic structures and gradient optimization is a flexible but less specialized technique for relaxed program induction. This paper unifies these two paradigms by leveraging *neural compilation and decompilation*: techniques for transforming code into neural network weights and transforming weights back into code [4, 5, 6, 7]. This hybrid algorithm retains both the generalization of program synthesis and the flexibility of gradient optimization.

The closest ideas to this paper are forms of *neurosymbolic programming* [3], and AutoML, which each mix elements of program synthesis, symbolic search, and differentiable computing [8, 9, 10, 11, 12, 13, 14, 15, 16, 17]. However, program synthesis combined with neural compilation and optimization is a unique and direct form of hybrid discrete-continuous neurosymbolic search.

**Neural Compilation**  The neural compilation algorithm in this paper is a replication of [6]. Historically, [4] established the Turing completeness of neural networks, which implies the existence of a neural compiler: a function that maps any Turing-complete program into the weights of a neural network. Shortly after, [5] created the first neural compiler, based on Pascal. However, this neural compiler could not tune compiled programs using gradient descent. Accordingly, [6] created the first neural compiler which was *adaptive* and could be locally tuned with gradient descent. This focused on a minimal assembly language that ran on a minimal differentiable computer, a type of recurrent neural network with explicit memory and addressing schemes. Afterward, [7] created a neural interpreter for a higher-level language called forth, which used a differentiable stack machine. However, both [6] and [7] utilized human-written programs as initializations for optimization. In contrast, this paper utilizes program synthesis as a method for efficiently finding abstract algorithmic structures.

Submitted to 36th Conference on Neural Information Processing Systems (NeurIPS 2022). Do not distribute.

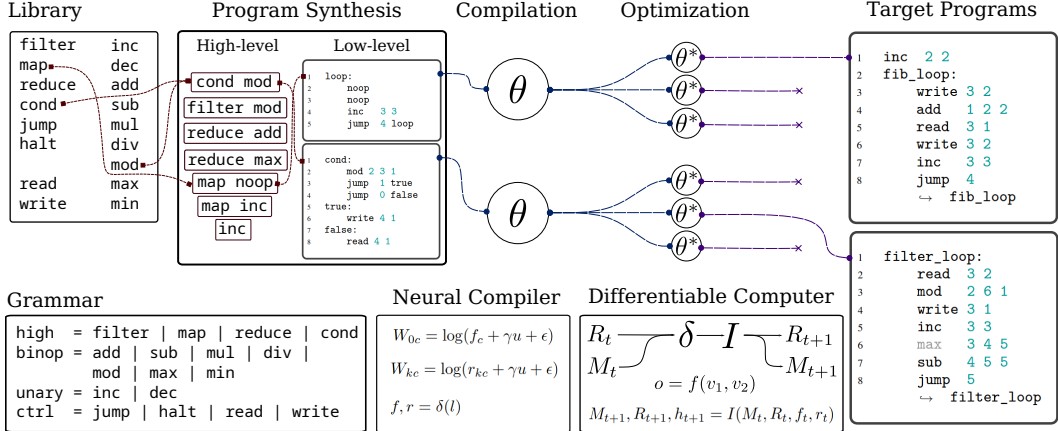

Figure 1: A neurosymbolic program induction algorithm which leverages program synthesis to find abstract algorithmic structure that is compiled into weights and optimized to find specific programs

**Program Synthesis** Program synthesis was anticipated as early as Turing, and underlied Solomonoff's theory of inductive inference [18, 19], though the first critical historical milestone for program synthesis was the FlashFill program [20]. Components such as version space algebras, equivalence-Graphs, and synthesis through unification were essential in efficiently searching the combinatorial space of computer programs, which is otherwise intractable [1, 21, 22, 23, 24]. However, an even more critical feature of effective program synthesis is *abstraction*: the ability to create a customized library of higher-level programs which capture common patterns, making program induction more efficient [25, 26, 27, 28]. Finding abstractions and composing them is a central feature of human intelligence, and therefore also central for machines [29, 30, 31, 32, 33].

**Differentiable Computing** The minimal differentiable computer in this paper builds on recurrent neural networks [34] and their many extensions [35, 36, 37, 38, 39]. These architectures aim to induce programs from data via gradient optimization. However, doing so is challenging due to an overabundance of local but suboptimal solutions, as well as technical issues such as unstable gradients [40]. Despite this, with the correct formulation and resources, it is possible to recover interesting algorithms, such as simple planning or sorting algorithms [37].

## 2 Algorithm

The algorithm in this paper (depicted in Figure 1) has three primary steps: synthesis, compilation, and optimization. First, a program synthesis algorithm searches a high-level language for abstract program templates. Then, these programs are mapped into the minimal assembly language specified in [6]. A neural compiler converts this lower-level assembly program into the weights of a neural network, which then act as the initialization for optimization. Finally, the optimization algorithm tunes this program by using gradient descent. Optimization is repeated multiple times, as algorithmic induction is highly sensitive to initialization.

Minimal assembly language acts as a common interface between synthesis and optimization, and neural compilation is the means for using this interface. While minimal assembly is easily compiled, it is unideal for direct program synthesis, even when using equivalence graphs or version space algebras. For example, in a 4 register machine with 13 instructions, there are roughly 100 million distinct instruction-argument pairs in only three lines of minimal assembly. Because of this, designing an appropriate high-level language plays a large part in the success of the overall algorithm, especially since the language design controls what abstract algorithm templates are included. By carefully manipulating this, it should be possible to recover desirable target programs reliably.

## 2.1 Neural Compilation

Fundamentally, the minimal differentiable computer is a recurrent neural network with a controller $\delta$, interpreter $I$, language $L$, memory tensor $M$, register tensor $R$, and halt state $h$: $(\delta, I, L, M, R, h)$. The controller outputs an instruction $f$ and register arguments $r$ based on the instruction register $l$:

$$f, r = \delta(l) \tag{1}$$

Where $f$ is a one-hot encoding corresponding to an assembly instruction, and $r$ contains three one-hot encodings specifying register arguments. Typically the first two registers are inputs, and the third register is used to store output, such as $\mathtt{add}(r_1,\ r_2,\ r_3)$, which adds the values in $r_1$ and $r_2$ and stores the result in $r_3$. The controller is parameterized with four weight matrices $W_k$, which determine the function $f$, and the arguments $r$ as functions of $l$, the instruction register:

$$f = \mathtt{softmax}(W_0 l) \qquad a_k = \mathtt{softmax}(W_k l) \quad k > 0 \tag{2}$$

Neural compilation works by inverting $\mathtt{softmax}$ and setting $W_k$ to produce a desired instruction $(f, r)$ at instruction count $c$. Uniform noise $u$ is added with a magnitude $\gamma$, which allows flexible optimization but preserves the desired instruction. A small constant $\epsilon$ is added for numerical stability:

$$W_{0c} = \log(f_c + \gamma u + \epsilon) \qquad W_{kc} = \log(r_{kc} + \gamma u + \epsilon) \quad k > 0 \tag{3}$$

The machine interpreter $I$ is a function that uses the recurrent state (memory and registers) and the instruction specified by the controller:

$$M_{t+1}, R_{t+1}, h_{t+1} = I(M_t, R_t, f_t, r_t) \tag{4}$$

First, arguments $r_{kt}$ are resolved to their values $v_{kt}$ by a register lookup:

$$v_{kt} = r_{kt} R_t \tag{5}$$

Many functions, such as $\mathtt{add}$ only depend on input registers, and not on memory state. For a machine in base $b$, outputs are stored in a $|L| \times b \times b \times b$ lookup table $T$, where the first dimension corresponds to a function $f$, the second two dimensions represent values $v_1$ and $v_2$, and the final dimension encodes the output of $f(v_1, v_2)$. For the $\mathtt{read}$ instruction, the $b \times \mathbf{1} \times b$ sub-tensor of $T$ corresponding to reading is set to the current memory, $M_t$, and for special instructions $\mathtt{write}$, $\mathtt{jump}$, $\mathtt{halt}$, sub-tensors of $T$ are zero. $T$ is indexed differentiably using an Einstein summation, which is analogous to using an addition or multiplication table, but for all assembly instructions and arguments

$$o_t = \mathtt{einsum}(klmn, k, l, m \to n, T, f_t, v_{1t}, v_{2t}) \tag{6}$$

Then, registers are updated with a soft write parameterized by $r_3$, the output argument:

$$R_{t+1} = R_t \odot (1 - r_{3t}) + o_t \otimes r_{3t} \tag{7}$$

Writing to memory uses $w_t$, the scalar component of $f_t$ representing the write probability.

$$M_{t+1} = (1 - w_t)M_t + w(1 - v_1 t) \cdot \mathbf{1} \odot M_t + v_{1t} \odot v_{2t} \tag{8}$$

The jump instruction modifies the instruction register $l$ probabilistically using $j$, a scalar component of $f_t$ representing the jump probability, and $z$, the scalar component of $v_{1t}$ representing the probability that $v_{1t}$ is zero. $T_{\mathtt{inc}}$ denotes the sub-tensor of $T$ for the increment instruction, and $l_n$ would be the next instruction if the jump is not taken.

$$l_n = l_t \cdot T_{\mathtt{inc}} \qquad l_{t+1} = l_n(1 - j) + r_{2t}z + jl_n(1 - z) \tag{9}$$

Finally, the halting probability $h_t$ is simply a scalar component of $f_t$.

## 2.2 Optimization

Once a program has been compiled into program weights, it is optimized using the adam optimizer [41], and a loss function with two components: correctness and efficiency. Correctness is a masked cross-entropy loss between a predicted tensor $P$ and labels $L$ across the final dimension. $\mu$ is a vector mask across the first dimension. Correctness is calculated for registers $R$ and memory $M$:

$$\underbrace{\mathcal{L}(P, L, \mu) = \mu \odot \mathtt{cross\_entropy}(P, L)}_{\text{correctness}} \tag{10}$$

Efficiency is a differentiable penalty for the number of computation steps:

$$h_{t>k} = \mathtt{max}(h_{t \leq k}) \qquad \underbrace{\mathcal{L}(h_t) = \sum \mathbf{1} - h_t}_{\text{efficiency}} \tag{11}$$

And the composite loss is a weighted combination of the correctness and efficiency losses:

$$\underbrace{\mathcal{L}(\hat{M}, \hat{R}, M, R, h, \mu)}_{\text{composite}} = \lambda(\underbrace{\mathcal{L}(\hat{M}, M, \mu_M) + \mathcal{L}(\hat{R}, R, \mu_R)}_{\text{correctness}}) + \lambda \underbrace{\mathcal{L}(h)}_{\text{efficiency}} \tag{12}$$

Neural networks and optimization components are implemented in $\mathtt{jax}$ and $\mathtt{equinox}$ [16, 17].

```
1   map_loop:
2       read   1 2
3       inc    2 2
4       write  1 2
5       inc    1 1
6       jump   3
    ↪   map_loop
7
8
9
10
```

```
1    inc  4 4
2    inc  4 4
3    sum_loop:
4        read   3 2
5        add    1 2 1
6        inc    3 3
7        max    3 4 5
8        sub    4 5 5
9        jump   5
     ↪   sum_loop
10   write 3 1
```

```
1    inc   2 2
2    fib_loop:
3        write  3 2
4        add    1 2 2
5        read   3 1
6        write  3 2
7        inc    3 3
8        jump   4
     ↪   fib_loop
9
10
```

Listing 1: Minimal assembly code for map, sum-reduce, and fibonacci functions

## 3 Experiments

These experiments explore which algorithms can be recovered via program synthesis, optimization, or a combined algorithm. An ideal evaluation task involves high-level algorithmic structure that can be established via program synthesis but contains sub-components that are continuous or best optimized as neural networks. Program synthesis finds the overall structure of a program, and the local optimizer tunes this program locally. The primary experiment uses a budget of $k = 100$ optimization runs and compares structured initializations to random initializations. Since many algorithms share a common structure (recursion, looping, conditionals, etc), starting with an algorithm template acts as a positive inductive bias, similar to how the choice of network architecture affects program behavior. Recovery is based on observational equivalence over a dataset of sampled program outputs. This allows recovering syntactically different solutions to a problem and discourages overfitting to a particular input-output pair.

Generally, algorithmic skeletons are better initializations than random initialization, but it is common for differentiable tuning to discard large parts of algorithm structure in certain problems. Since program synthesis finds various algorithmic skeletons, it outperforms using multiple uniform random initializations. Even programs that aren't directly enumerated, such as the Fibonacci program (Listing 1, Table 1), can be recovered using the combination of synthesis and tuning. Introducing no-ops into program synthesis (and not penalizing them) can be advantageous, as gradient descent tuning does not naturally model concepts like insertion. Table 1 includes no-op-padded program initializations in the second half. Interestingly, a few results defy intuition, such as that `inc` is harder to find, and that `map dec` is not transitive with `map inc`, we hypothesize that this is because it is difficult for optimization to represent simpler programs, as it typically saturates the available instructions. A preliminary grid search found a noise parameter in the neighborhood of $\gamma = 0.3$, which is sufficient for gradient information to capture the local program space.

Table 1: Recovery rates for selected algorithms and initializations

| Algorithm | inc | map inc | map dec | reduce | Parity | Fibonacci |
|---|---|---|---|---|---|---|
| Optimization | 19% | 86% | 56% | 41% | 95% | 4% |
| Synthesis | 100% | 100% | 100% | 100% | 100% | **0%** |
| Both | 100% | 100% | 100% | 100% | 100% | **75%** |
| Initializations | | | | | | |
| `map inc` | - | 100% | 7% | - | 7% | 26% |
| `map dec` | - | 45% | 100% | - | 6% | 41% |
| `loop no-op` | 100% | 100% | 100% | 100% | 49% | **75%** |

Table 1 shows the percent of perfect algorithms recovered for each algorithm and different initial program structures. Program synthesis will recover many of the program structures listed in this table, some of which will be near-misses to a desired program. Then, differentiable tuning can find a local variant of the program that is close to a desired program. This shows that, for this neural architecture, the combined synthesis-compilation algorithm is more computationally efficient than optimization alone.

## 4 Limitations & Future Work

While the neural compilation method introduced by [6] is straightforward to compute and implement, it could be more adaptive and general. One major limitation is the lack of parameters in the network model: each instruction and its arguments are determined only from the instruction register, and the function used is linear with a softmax activation. For example, in a network model for a 32 instruction program, there are only $3,640$ parameters. While this is desirable for some applications, it is in contrast to implementations such as [34, 35, 36, 37] where network behavior is a function of memory and input, and modern network architectures that have millions or billions of parameters. Also, using a recurrent neural network inherently makes representing long programs and sequences difficult because of the unstable gradient problem. Future work will explore neural compilation techniques that are more adaptive and tunable but retain interpretability.

The minimal differentiable computer introduced in [6] is a relatively weak program induction baseline. Future work will include stronger end-to-end differentiable algorithm induction baselines, especially modern architectures [37, 39]. However, the minimal differentiable computer is highly compute and parameter efficient.

The program synthesis algorithm given in this paper is relatively simple compared to modern techniques. In particular, it does not generate abstractions or utilize neural search heuristics such as those in [25]. These elements are modular and would most likely boost performance, especially if used in tandem with differentiability-based tuning. Finally, given sufficient computing power and time, a more advanced version of this algorithm would likely be successful on more interesting tasks, such as sorting or planning algorithms that are embedded in larger neural programs.

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

# A  Appendix

## A.1  Interpreting distributed programs

Differentiable programs are *distributed*, because the instruction register is a probability distribution. This means that multiple instructions can be carried out at once, which makes one-to-one decompilation difficult, and also prevents these programs from being easily human-interpretable. Distributed execution is affected by two factors in our model: the probability that a current instruction is a jump instruction, and the probability that the comparison register for the jump instruction is equal to zero. Also, every operation is distributed, so each register's values and all memory values are multinomial distributions created by softmax, which overlap with one another. Thus, in a longer non-trivial program, decompiling network weights into a one-to-one interpretation is more difficult. However, starting with a decompilable algorithm increases the probability that a tuned algorithm will be interpretable, as the initialized algorithm is less distributed than a naturally recovered algorithm.

```
1    read   0 1
2    inc    1 1
3    read   1 1
4    write  0 1
5    halt
```

```
1    map_loop:
2        read   1 2
3        inc    2 2
4        write  1 2
5        inc    1 1
6        jump   3
     ↪   map_loop
```

```
1    cond:
2        mod   2 3 1
3        jump  1 true
4        jump  0 false
5    true:
6        write 4 1
7    false:
8        read  4 1
```

```
1    loop:
2        noop
3        noop
4        inc   3 3
5        jump  4 loop
```

```
1    filter_loop:
2        read   3 2
3        mod    2 6 1
4        write  3 1
5        inc    3 3
6        max    3 4 5
7        sub    4 5 5
8        jump   5
     ↪   filter_loop
```

```
1    inc   2 2
2    fib_loop:
3        write 3 2
4        add   1 2 2
5        read  3 1
6        write 3 2
7        inc   3 3
8        jump  4
     ↪   fib_loop
```