# OpenReview forum: "Synthesized Differentiable Programs"
_NeurIPS.cc/2022/Workshop/nCSI — nCSI WS @ NeurIPS 2022 Poster_

### Official Review · Reviewer_upK8 · 2022-10-14
**Very interesting approach to combining neural compilation with program synthesis**

**Rating:** 2
**Confidence:** 2

**Review:**

This work focuses on a super interesting idea of combining program synthesis with neural optimization. Specifically suggesting a novel algorithm that combines transforming explicit code into neural weights and transferring these back into code. Thus, representing a very interesting neuro-symbolic approach. The brief experiments provide a good proof of principle. Overall, I believe there are several relevant points within this work concerning this workshop that should contribute to fruitful discussions and exchange of ideas.
Having said this I found the paper a little difficult to understand which I attribute to the overall structure of the work. E.g. a clearer introduction and final conclusion and a stronger distinction between motivation, related works and how they differ from this approach would be beneficial. But also smoother transitions between the sections would be beneficial, especially as the page limit still allows for this.

---

### Official Review · Reviewer_8uFC · 2022-10-14
**Novel algorithm for neuro-symbolic program induction**

**Rating:** 2
**Confidence:** 2

**Review:**

Authors propose the novel technique for neuro-symbolic program induction. For this, they combine program synthesis, neural compilation and differentiable optimization together.

The novel algorithm offers interesting insights into one-to-one mapping between programs and their compilation as neural-network's weights. Since the neural-compilation probabilistic is, the process of decompilation is noisy as well. Nonetheless, the percent of perfectly recovered algorithms (Table 1) is impressive. It follows that the combination of synthesis and optimization allows for recovery of such algorithms as Fibonacci functions.

Despite the impressive results, it is difficult to judge the performance of the proposed algorithm since there are no comparisons with other works brought forward. It would largely improve the quality of the paper if the authors will add experimental settings where such comparison will be of at most importance.

Small remarks and wishes:
The described neural compilation process (Section 2.1) is detailed and insightful for the reader to understand the premise of the new algorithm. However, one cannot follow Eq. (8) and (9) since the meaning of the operators $\odot$ and $\otimes$ is not specified. Are these operators elementwise (scalar) or vectorwise (tensor) to be understood? The same can be stated about Eq. (10). Sadly, in Eq. (11) the source of the variable $h$ is not explained. Is it the halt state as described at the beginning of the subsection?

The references should be doubly checked. Some of them include a link exclusively to arXiv [13, 17, 29, 32, 33, 36, 37, 39, 41] and some are not complete [21, 26, 27]. It would be much better either replace the links to arXiv all together, or make the remark that some references are accessible exclusively as arXiv-preprints.

---

### Meta-Review · Area_Chair_KHEA · 2022-10-19

**Recommendation:** 2
**Confidence:** 3

**Metareview:**

The paper is good for the workshop. Please try to make the paper more readable, and going forward you need a better comparison with existing work.

---

### Decision · Program_Chairs · 2022-10-20

Accept (Poster)